# Binding Studies and Lead Generation of Pteridin-7(8*H*)-one Derivatives Targeting FLT3

**DOI:** 10.3390/ijms23147696

**Published:** 2022-07-12

**Authors:** Suparna Ghosh, Seung Joo Cho

**Affiliations:** 1Department of Biomedical Sciences, College of Medicine, Chosun University, Gwangju 501-759, Korea; s.ghosh@chosun.kr; 2Department of Cellular and Molecular Medicine, College of Medicine, Chosun University, Gwangju 501-759, Korea

**Keywords:** FMS-like tyrosine kinase-3, MM-PB/GBSA, structure–activity relationship, linear interaction energy, umbrella sampling, free energy perturbation, 3D-QSAR

## Abstract

Ligand modification by substituting chemical groups within the binding pocket is a popular strategy for kinase drug development. In this study, a series of pteridin-7(8*H*)-one derivatives targeting wild-type FMS-like tyrosine kinase-3 (FLT3) and its D835Y mutant (FL3_D835Y_) were studied using a combination of molecular modeling techniques, such as docking, molecular dynamics (MD), binding energy calculation, and three-dimensional quantitative structure-activity relationship (3D-QSAR) studies. We determined the protein–ligand binding affinity by employing molecular mechanics Poisson–Boltzmann/generalized Born surface area (MM-PB/GBSA), fast pulling ligand (FPL) simulation, linear interaction energy (LIE), umbrella sampling (US), and free energy perturbation (FEP) scoring functions. The structure–activity relationship (SAR) study was conducted using comparative molecular field analysis (CoMFA) and comparative molecular similarity indices analysis (CoMSIA), and the results were emphasized as a SAR scheme. In both the CoMFA and CoMSIA models, satisfactory correlation statistics were obtained between the observed and predicted inhibitory activity. The MD and SAR models were co-utilized to design several new compounds, and their inhibitory activities were anticipated using the CoMSIA model. The designed compounds with higher predicted pIC_50_ values than the most active compound were carried out for binding free energy evaluation to wild-type and mutant receptors using MM-PB/GBSA, LIE, and FEP methods.

## 1. Introduction

FLT3 is best described for its pivotal role in the constitutive activation and development of acute myeloid leukemia (AML) in humans [1]. It is primarily expressed in murine and hematopoietic stem cells and is responsible for the natural development of the immune system. Structurally, FLT3 consists of five immunoglobulin (Ig)-like extracellular domains, a single transmembrane (TM) domain, a juxtamembrane (JM) domain inside the cytoplasm, a cytoplasmic tyrosine kinase domain (TKD) separated by a kinase insert (KI), and a C-terminal intracellular domain (Appendix A) [2,3]. In the inactive state, FLT3 exists in its unbound, monomeric, and unphosphorylated forms. Upon binding to the indigenous ligand FL, FLT3 undergoes a conformational change. This conformational change occurs by unfolding the receptor and subsequent receptor–receptor homodimerization, bringing the kinase domain in proximity to the intracellular module, allowing the phosphorylation of the tyrosine residues (Y589, Y591, and Y599) in the JM domain. This leads to a cascade of phosphorylation and activation of secondary mediators, including STAT5, PI3K/Akt/mTOR, and Ras/Raf/MAPK oncogenic signal transduction (Appendix A) [1,4]. The premature activation of transcription factors triggers cell proliferation and impedes cell differentiation and apoptosis in leukemia cells. As simplified in Appendix A, the autoinhibited kinase domain (KD) consists of an N and C bi-lobal structure with an activation loop and a JM domain (PDB ID 6JQR) [5]. The interaction between the KD and JM domains prevented ATP binding. The N-lobe has an α-helix (αC-helix) and five antiparallel β-sheets, namely, β1–β5. The C-lobe, on the other hand, has seven α-helices and three β-sheets, namely, αD–αI and β6 to β8. The activation loop comprised two twisted β-sheets (β10 and β11). An additional β-sheet is present in the JM domain, termed as βJ2. The N and C lobes are connected by a polypeptide stretch, called the hinge loop, which allows the rotational movement of the two lobes relative to each other.

Active and inactive FLT3 kinase domains can be distinguished by their characteristic ‘DFG-in’ and ‘DFG-out’ configurations. The phenylalanine residue of the DFG motif flipped 180° from its active configuration to an inactive conformation facing the active site. This creates an additional hydrophobic pocket for type II inhibitors with an elongated geometry to interact with the residues in the αC-helix [6]. Several clinical and preclinical studies have found that mutations and the overexpression of FLT3 are associated with a poor prognosis of AML. One of the most common mutations, D835Y, has been detected in the kinase activation loop. Nevertheless, mutations in residues I836, D839, Y842, and the gatekeeper residue F691 adjacent to the active site pocket were also found in patients with a lower frequency [7,8]. Mutations in TKD consecutively activate tyrosine kinase, which phosphorylates the intracellular domain at various sites and recruits many cytoplasmic adapter proteins for protein–protein interactions with the FLT3 receptor.

As FLT3 involvement becomes more prevalent in oncogenic conditions, many small molecules that target FLT3 tyrosine kinase have been discovered. Midostaurin, sorafenib, and lestaurtinib were used as multikinase C inhibitors to improve clinical outcomes in patients with AML [9]. However, their antileukemic activities were limited when used as monotherapy, and adverse cytotoxicity was observed. Therefore, researchers are working to develop next-generation inhibitors that selectively target the FLT3 receptor [10]. Many of them are currently being evaluated in clinical trials and have higher potencies than multikinase inhibitors, such as gilteritinib, quizartinib, and crenolanib. Crenolanib and gilteritinib fall into the category of type I inhibitors and target both active and inactive kinase states, whereas quizartinib is a type II inhibitor specific to inactive-state conformations (Appendix A) [11]. On the other hand, type I inhibitors have an identical set of chemical interactions in the ATP pocket, forming one to three H-bond interactions similar to that of the adenine moiety of ATP molecules. In addition, they occupy the proximal A-loop or allosteric site, front pocket, and P loop, providing an additional selectivity property for the type I inhibitors compared to that of the type II inhibitors.

Computational drug design is a popular choice for discovering small molecules targeting kinase receptors. In our previous study [12], we performed the molecular modeling of pyrimidine4,6-diamine derivatives against inactive FLT3 as type II inhibitors. In this paper, we conduct the modeling study of 35 pteridin-7(8*H*)-one compounds as type I inhibitors targeting the active FLT3 conformer. The compounds exhibited a wide range of inhibitory activity (pIC_50_ 5.26–8.80) against FLT3, which was studied by Sun et al. [13]. The molecular docking and MD simulation studies of the most active compound C31 were conducted together with other structurally diversified compounds from the dataset to study the critical interactions and binding stability of the complexes. We conducted MM-PB/GBSA, FPL, LIE, and FEP calculations to evaluate protein–ligand binding affinity and build the scoring models. We also applied US methods to estimate the effective binding free energy of C31 in complexes with wild-type and mutant receptors through the unbinding pathway. The last 1 ns average MD structure of C31 was retrieved to develop the CoMFA and CoMSIA models for the SAR study. Several new compounds were designed, which were further investigated for inhibitory activity prediction using the CoMSIA model. The binding affinities of the newly designed compounds were evaluated by MM-PB/GBSA, LIE, and FEP calculations.

## 2. Results and Discussion

The optimal ligand-binding orientation at the active site was predicted using the molecular docking study. Docking pose verification is a critical step since it is used in molecular simulation, MM-PB/GBSA binding energy evaluation, and lastly, the generation of 3D-QSAR models. The 2D structure of pteridin-7(8*H*)-one-based compounds that have been chosen for molecular docking studies is shown in Figure 1a. None of the compounds in the dataset had FLT3 co-crystal forms; the most active compound C31 was cross-docked into the FLT3 binding pocket, which the gilteritinib molecule had previously occupied. The top-scoring ligand poses in a cross-docking experiment are not always accurate for evaluating the final docking result. Therefore, in addition to the top-scoring solution, we employed RMSD evaluation between the docked pose and gilteritinib crystal pose by the LigRMSD web server [14]. According to this study [15], RMSDs between the docked and crystal poses of 2.0–3.0 Å are an acceptable docking solution. The final docked complex should also comply with ECIDALs norms, in which the essential chemical interaction and binding configuration of the analogous ligands were thoroughly inspected across the same protein family from the PDB database. The docking interaction of the top-ranked FLT3-C31 within the binding pocket is shown in Figure 2b. Compound C31 and gilteritinib are designed to target active FLT3 in a type I inhibitor-like manner. The RMSD between the docked ligand and crystal ligand gilteritinib was found to be 2.2 Å. In addition, compounds C31 and gilteritinib have 2-phenylaminopyrimidine and 2-phenylaminopyrazine moieties in their structure, which share an identical chemical scaffold and form a bidentate H-bond interaction with the residue C694 at the hinge loop. C31 formed two additional H-bond interactions with residue L616 and the keto (-C=O) group of the DFG residue F830. Other notable interactions, such as π–π stacking and π–sigma interactions, were formed between the pteridine ring and residues Y693 and L818, respectively. Residues V624, A642, V675, and C828 formed hydrophobic interactions with the ligand. The comprehensive docking results were summarized in Appendix A, and the interaction diagrams were illustrated in Appendix A. The RMSDs with giltertinib for compounds C03, C06, C17, C22, and C28 were found to be less than 3.0 Å, indicating reasonable docking accuracy. Overall, the docking analysis suggested a satisfactory docking solution that could be utilized in the binding study of new compounds.

Since protein–ligand interaction is a highly thermodynamic process, a single docking experiment has several limitations. In the cross-docking experiment, the binding site was treated as a rigid body, and neither side-chain nor backbone movements were taken into account. Additionally, in the empirical scoring functions, water-mediated hydrogen bonding, de-solvation, and estimation of ligand binding energy by water swapping remain challenging. In many cases, the docked pose is not stable under physiological conditions. Thus, MD simulation studies were employed to validate the docking solutions and the overall stability of the protein–ligand complexes. We manually mutated the residue D835Y in the FLT3 (Figure 1c) structure to make the FLT3_D835Y_–C31 complex. Therefore, eight total protein–ligand systems, i.e., FLT3–C31, FLT3_D835Y_–C31, FLT3–C01, FLT3–C03, FLT3–C06, FLT3–C17, FLT3–C17, and FLT3–C28, were subjected to 100 ns production simulations. The oscillation of the backbone αC of proteins and the heavy atoms of the ligands are plotted with respect to the simulation time in Figure 1d–k. The RMSD plots stated that the systems were converged within the first 20 ns of the simulations. The RMSDs of the protein and ligand were in the range of 1.0–3.5 Å. The FLT3 complexes with compounds C31, C01, and C17 were stable after initial convergence, although the RMSDs of FLT3–C06 and FLT3–C28 suggested multiple state conversions during the MD runs.

The MM-PB/GBSA binding energy is a frequently used method to calculate the end-state binding free energy between protein–ligand complexes. We collected the last 2 ns trajectory or 200 snapshots from each system to compute the MM-PB/GBSA binding free energy. The entropy term (TΔS) was calculated from the last 80 snapshots of the 2 ns trajectory and then subtracted from the ΔTOTAL term to obtain the final MM-PB/GBSA binding energy. The comprehensive assessments of the MM-PB/GBSA binding energy terms are shown in Appendix A. The final ${\Delta G}_{\mathrm{MM}-\mathrm{PB}/\mathrm{GBSA}}$ values were estimated to be −32.15 kcal/mol and −30.54 kcal/mol for the C31-bound wild-type and mutant FLT3 complexes, respectively.

The binding energies for the compounds C01, C03, C06, C17, C22, and C28 with wild-type FLT3 were found to be −22.70 kcal/mol, −22.62 kcal/mol, −21.71 kcal/mol, −26.84 kcal/mol, −30.83 kcal/mol, and −30.97 kcal/mol, respectively. Subsequently, we computed the residue-specific binding energy contribution within the 4.0 Å distance from the ligand atoms. The residues K614, L616, G617, V624, A642, E692, Y693, C694, L818, and F830 were found to be the major binding energy contributors in MM-PB/GBSA terms. The residue-specific binding energy decomposition analysis is summarized in Appendix A, and the graphical illustration is shown in Figure 1m.

FPL simulation along with the unbinding pathway was conducted, which is based on the SMD principle. It is also a relatively straightforward approach for estimating the binding affinity between protein–ligand complexes. In this method, the ligands were forced to dissociate from the center of mass (COM) distance of the DFG residues through the caver-predicted unbinding tunnels (Appendix A) at a distance of about 5 nm toward the *X*-axis (Figure 2a). Initially (T = 0 ps), the pulling force was minimal, and the ligand was bound to the active site cavity, referred to as the *bound* state. Over the simulation, the pulling force was gradually increased until the ligands began to dissociate from the binding pocket. At that time (T = $T_{\max}$), the pulling force reached its peak, the ligand was separated from the cavity and mobilized into the solvent, termed rupture force ($F_{\max}$). The external force abruptly decreased and maintained a consistent plateau, referred to as the *unbound* state. Theoretically, the ligand with higher inhibitory activity poses a higher relative binding affinity. Thus, $F_{\max}$ could be applied to rank the inhibitor compounds. The external pulling forces and separation distances over time are shown in Figure 2b. The average F_max_ values for compounds C01, C03, C06, C17, C22, C28, and C31 were estimated to be 221.40, pN, 441.13 pN, 391.61 pN, 428.75 pN, 475.17 pN, 441.13 pN and 537.07 pN, respectively. In contrast, a lower $F_{\max}$ value ($F_{\max}$ = 453.51 pN) was obtained for the FLT3_D835Y_–C31 complex compared to the C22 and C31 systems.

Next, we calculated the LIE approximation over the two quasi-equilibrium states (*bound* and *unbound*) by computing the van der Waals and electrostatic interactions. For the compounds C01, C03, C06, C17, C22, C28, and C31, the absolute binding energies using the LIE approximation were determined to be −28.76 kcal/mol, −27.18 kcal/mol, −28.71 kcal/mol, −30.65 kcal/mol, −30.97 kcal/mol, −30.35 kcal/mol, and −28.92 kcal/mol, respectively. In comparison to the wild-type FLT3, compound C31 had rather lower absolute binding free energy (${\Delta G}_{\mathrm{LIE}}$ = −28.17 kcal/mol) to the mutant receptor. In-depth $F_{\max}$ and LIE calculations are summarized in Appendix A. Following that, the US simulation was applied to the C31-bound FLT3 and FLT3_D835Y_ complexes to evaluate the effective binding free energy profile along with their dissociation pathway. A total of 25 evenly distributed overlapping windows were extracted from the FPL trajectories and used for biased sampling simulations. The binding free energy and sufficient sampling could be traced by analyzing the PMF curve and the umbrella histogram with respect to the reaction coordinates (ξ), as shown in Figure 2c–f. In PMF, the free energy began from zero and then dropped to a minimum value. Subsequently, the energies were gradually increased to attain a stable value, where non-covalent interactions between protein and ligands were completely broken. The binding free energies from the US simulation ($\Delta G_{\mathrm{US}}$) of the most active compound C31 for wild-type and mutant FLT3 were calculated to be −10.73 ± 1.27 kcal/mol and −9.49 ± 0.57 kcal/mol, respectively. The convergence of the calculation could be validated by the histogram profiles of overlapping neighboring windows.

The FEP simulation was performed with the last 1 ns average MD structure of the protein–ligand complexes. The vdW and coulombic interactions of the ligands were sequentially turned-on in the solute in a complex and isolated form by alter-λ simulations. The energy convergence plots in the forward and reverse directions are shown in Appendix A. The first 40% of the trajectory data were discarded to eliminate any convergence error. The remaining data were used to calculate the different binding energy terms in the calculation of the FEP and are summarized in Appendix A. The final absolute binding free energy ($\Delta G_{\mathrm{FEP}}$) values from the FEP simulation were determined to be −14.83 kcal/mol, −14.64 kcal/mol, −13.68 kcal/mol, −16.77 kcal/mol, −15.04 kcal/mol, −17.61 kcal/mol, and −17.87 kcal/mol for compounds C01, C03, C06, C17, C22, C28, and C31, respectively. Compared to the wild-type complex, the FLT3_D835Y_–C31 complex exhibited lower binding free energy ($\Delta G_{\mathrm{FEP}}$ = −16.82 kcal/mol) in the FEP calculation.

Table 1 further emphasizes the final binding free energies of the protein–ligand complexes, which are derived using MM-PB/GBSA, FPL, LIE, and FEP methods. The experimental binding energies ($\Delta G_{\mathrm{EXP}}$) of the compounds were deduced from their inhibitory activity (IC_50_) and attempted to correlate with the computed binding free energies. During the correlation analysis, the binding energies of the FLT3_D835_–C31 complexes were ignored. The correlation plots between the experimental binding energies and computed binding energies of the seven compounds are shown in Figure 3. A good correlation coefficient ($R_{\mathrm{MM}-\mathrm{PB}/\mathrm{GBSA}}$ = 0.92) was obtained between the $\Delta G_{\mathrm{EXP}}$ and ${\Delta G}_{\mathrm{MM}-\mathrm{PB}/\mathrm{GBSA}}$. However, the binding energies were overestimated by the MM-PB/GBSA method. In the FPL model, the $F_{\max}$ values are poorly correlated with the $\Delta G_{\mathrm{EXP}}$ values ($R_{F_{\max}}$ = −0.55). The correlation coefficient ($R_{\mathrm{LIE}}$) between $\Delta G_{\mathrm{EXP}}$ and ${\Delta G}_{\mathrm{LIE}}$ was calculated to be 0.60. Thus, the observations suggested a significant limitation and required special attention when utilizing the FPL and LIE models for ligand ranking. Although, both models were able to distinguish the binding affinity differences between C31 and FLT3 variants. In the FEP model, the correlation coefficient ($R_{\mathrm{FEP}}$) between $\Delta G_{\mathrm{EXP}}$ and ${\Delta G}_{\mathrm{FEP}}$ was estimated to be 0.71, which is statistically reasonable and could be utilized to assess the binding affinities of unknown compounds.

Appendix A represent the dataset compounds and their molecular alignments on C31. The statistical analysis of CoMFA and CoMSIA is summarized in Table 2. The acceptable parameters of each statistical term are listed in the ‘Threshold values’ column based on the previously published literature. In CoMFA analysis, q^2^ and r^2^ were obtained as 0.768 and 0.982, greater than 0.5 and 0.6, respectively, at the ONC of 3. The steric and electrostatic contributions of the CoMFA scheme were found to be 54.2% and 45.8%, respectively. To generate the best statistically significant CoMSIA model, we used five descriptor fields, such as steric (S), electrostatic (E), hydrophobic (H), H-bond donor (D), and H-bond acceptor (A) in the permutation-combination process as shown in Appendix A. The best q^2^ and r^2^ values of 0.844 and 0.972 were obtained in SH combination at ONC of 4, and, therefore, SH was selected as the final CoMSIA model. The final contributions of the steric and hydrophobic fields were found to be 46.8% and 53.2%, respectively. However, any QSAR models are insufficient without being externally validated by test set compounds that were not used during model development. Thus, the external validation was conducted by estimating the predictive correlation coefficient or $r_{\mathrm{pred}}^{2}$. In CoMFA and CoMSIA, the values of $r_{\mathrm{pred}}^{2}$ were determined to be 0.919 and 0.918, respectively, greater than the constrained value of 0.6, signifying that both models were statistically reliable and had good predictability. The predicted activity of the dataset compounds, which includes both training and test set compounds from CoMFA and CoMSIA studies are reported in Appendix A. Figure 4a,e represent the PLS regression plots between the observed and predicted activity of compounds from the CoMFA and CoMSIA models. Moreover, we also calculated other statistical parameters, such as $r_{m}^{2}$ or ${r^{\prime }}_{m}^{2}$, $Q_{\mathrm{Fn}}^{2}$ (n = 1, 2, 3), and $Q_{\mathrm{ccc}}^{2}$ matrices, all of which were calculated to be within the well-accepted parameters.

Next, we conducted the applicability domain analysis to visually detect the outliers [16]. It is a theoretical chemical space in which the QSAR model could reliably predict the descriptor properties of compounds. The AD analysis of CoMFA and CoMSIA is illustrated in Figure 4b,f by the distance-based Williams plot. The standardized residual values of the training set and test set compounds were plotted against their leverage values within a square area of σ = ±3 and warning leverage (h*). Compounds with a leverage value greater than h* were considered outliers and significantly affected the regression slope of the QSAR models. In our study, none of the compounds were outside the warning leverage (h* = 0.29), suggesting the robustness of the 3D-QSAR models.

The contour maps analysis from the 3D-QSAR study was conducted to explore the favorable and unfavorable sites for chemical substitution. As shown in Figure 4c–h, the field effects of the chemical descriptors from CoMFA and CoMSIA were graphically represented by contour polyhedrons around the C31-bound active site. In both CoMFA and CoMSIA, two green contours appeared at the R_1_ and R_3_ positions near the solvent-exposed area of the active sites, indicating that the presence of bulky steric chemical groups in that region could increase the inhibitor potency. In contrast, a large yellow contour appears near the DFG residues, suggesting a disfavored substitution for bulky steric groups at that position. Compounds C04, C07, C08, C09, and C12 non-steric groups in their R_1_ and R_3_ positions exhibited lower inhibitory activity (pIC_50_ < 0.7) than the other dataset compounds.

On the other hand, compounds C31 and C32 consist of steric groups, such as methyl (-CH_3_) or methoxy (-OCH_3_) groups, in the *meta*-position instead of the *para*-position of the piperazine moiety, which allocated them in proximity to the green contours. It might favor the critical inhibitory potency of these two highest active compounds. The blue and red contours (Figure 4d) suggested a favorable substitution for the electropositive and electronegative chemical groups. In that chemical space, compounds with positively charged nitrogen (N atoms) or amine (-NH_2_) groups might enhance the inhibitory activity against FLT3. An orange contour near the aniline moiety at the R_2_ position towards the residue F830 suggested that a small hydrophobic substitution could be favorable (Figure 4h). Taken together, the overall observation was emphasized as a SAR scheme in Figure 5a.

In the context of SAR, we initiated the inhibitor design using substitution growth methods. The contour maps suggested a large steric substitution in the R_1_ and R_3_ positions, although this is not infinite. The addition of substantially bulky chemical components may result in steric clash and failure of ligand insertion into the binding pocket. Moreover, the designed compounds should satisfy Lipinski’s criterion and have a low complexity in the chemical synthesis route. Similarly, the compounds should be designed with scaffolds similar to the dataset compounds. A rather heterogeneous molecule may not be adequately evaluated by a 3D-QSAR model, causing it to be assigned outside the applicability domain or the chemical space. Earlier modeling studies have reported that D835 mutations alter the conformational changes of the phenylalanine residue (F830) of the conserved DFG motif, affecting the vdW and electrostatic interactions, which influence the binding affinity regardless of the type I or type II inhibitors [17,18]. Our multiple binding energy computation schemes estimated a lower binding affinity of the most active compound to the mutant receptor. Therefore, growing molecular probes from the R_2_ position towards DFG residues may contribute to additional steric or electrostatic interactions and ultimately improve the binding affinity of the designed compounds. This could be reinforced from the SAR study, as we obtained that the non-steric, hydrophobic, and electronegative groups could be favorable for improving the inhibitory potency of C31. By considering the above factors, we designed up to 50 new compounds (Appendix A), and their activity was predicted by the CoMSIA model. We introduced the steric substitution to the R_3_ position as a first step while leaving the other positions unaltered. At the R_2_ site, we added hydrophobic and electronegative chemical entities (-C=O, -CF_3_), while the R_3_ position remained unchanged. Following that, chemical probes were grown in the R_1_ position, with varying degrees of substitution in the R_2_ and R_3_ positions. Beyond the SAR scheme, we also incorporated investigational electronegative groups, such as chlorine and fluorine, as probe moieties. Thirteen designed compounds, namely, D02, D03, D04, D15, D16, D17, D18, D25, D26, D27, D45, D46, and D47, were predicted to have higher pIC_50_ values than the most active compound (Figure 5b). The binding affinities of these 13 compounds were performed using the MM-PB/GBSA, LIE, and FEP methods targeting wild-type FLT3 and D835Y mutant. RMSD plots of the wild-type and mutant complexes in a complex with the designed compounds are shown in Appendix A. The last two ns snapshots were extracted from the MD trajectories to calculate the MM-PB/GBSA binding free energy (Appendix A). Compounds D03, D15, D17–18, D25–26, and D46–47 had higher binding free energies than C31 in complexes with wild-type receptors. In contrast, D02, D04, D15, D18, D25–26, and D46–47 exhibited higher binding free energies in complexes with the mutant receptor. The last 1 ns average MD complexes of the designed compounds were employed for the FPL and FEP simulation studies. The potential mean force and displacement distance of the ligands over the simulation time are shown in Appendix A. The calculated LIE terms for the wild-type and mutant complexes are shown in Appendix A. Compounds D03, D04, D15–16, D27, and D46–47 had higher binding free energies in complexes with FLT3 receptors than C31. The FEP convergence plots of the designed compounds in complex with FLT3 wild-type and mutant variants are illustrated in Appendix A. The first 40% of the data is eliminated during the final FEP energy calculations to avoid the convergence error, as shown in Appendix A. The compounds D02, D04, D15, D18, D27, and D46–47 were shown to have stronger affinity for FLT3 receptors than C31. In Figure 6, the computed binding free energies from the MM-PB/GBSA, LIE, and FEP models are compared. The designed compounds with higher binding free energies than the most active compounds are designated by asterisks.

## 3. Methodology

### 3.1. Structure Preparation and Molecular Docking

The Surflex-Dock module in Sybyl X 2.1 (Tripos Inc., St. Louis, MO, USA) was used to perform the molecular docking study. Before the docking experiment, the FLT3 crystal (PDB ID 6JQR) with a resolution of 2.20 Å was retrieved from the RCSB protein databank. Water molecules, solvent ligands, and co-crystallized ligands were removed from the protein structure. Missing residues K634-G636, K649-A650, and G831-I836 were modeled using the web version of the MODELLER webserver (University of San Francisco, San Francisco, CA, USA) in Chimera-1.14 (RBVI, UCSF, San Francisco, CA, USA). The residue S711-L780 or KI domain was excluded during model development. The final model was selected based on the lowest DOPE score and Ramachandran plot from the PROCHECK (DOE-MBI service, UCLA) analysis (Appendix A). Compound C31 was the most active compound in the dataset and selected as the representative candidate for the docking study. To prepare the 3D structure of C31, Sybyl X 2.1 was used to sketch, minimize, and assign gasteiger charges as described in earlier studies [19,20]. The receptor was prepared using the structure preparation tool performed with the Amber7 99 force field. The docking cavity was defined using protomol generation, in which the gilteritinib-bound position was used as the reference. Finally, the docking score between the receptor and C31 was calculated using the empirical Hammerhead scoring function. Several parameters, such as polar, hydrophobic, repulsive, solvation, and entropy terms, were considered during the scoring assignment in Surflex-Dock. The final docking score was expressed in terms of −logK_d_ units, where K_d_ stands for the dissociation constant of the ligand. The docking protocol was repeated for the compounds C01, C03, C06, C17, C22, and C28.

### 3.2. MD Simulation

MD simulations were performed in GROMACS 2019.5 using the Amber14sb force field, according to our previous studies [21,22]. The topology and parameter files of the ligands were generated using ACEPYPE (or AnteChamber PYthon Parser interfacE) [23] with gasteiger charges. The complexes were placed in a cubic periodic box and solvated using the TIP3P water model. The minimum thickness of the water wall was maintained at ~10 Å from the protein atoms. Adequate amounts of Na^+^ and Cl^−^ were added to neutralize the system and bring the salt concentration to 150 mM. Next, each system was energy minimized by the steepest descent integrator followed by a 200 ps constant volume ensemble (NVT) to achieve a temperature of 300 K and 400 ps constant pressure ensemble (NPT) to achieve a pressure of 1 bar using the modified Berendsen (V-rescale) thermostat and barostat algorithms. During the NVT and NPT runs, the protein backbone and the heavy atoms of the ligand were restrained. Thereafter, the systems were subjected to 100 ns of MD production run by removing the backbone restraint. Particle mesh Ewald (PME) and LINCS algorithms with a cut-off value of 12 Å were employed to control the electrostatic interaction and bond length constraints. Protein and ligand RMSDs were calculated using the built-in ‘*gmx rms’* function in gromacs.

### 3.3. MM-PB/GBSA Binding Energy Calculation

According to our previous research [21], the gmx_MMPBSA package [24] was used to calculate the various MM-PB/GBSA terms. The binding energy (${\Delta G}_{\mathrm{MM}-\mathrm{PB}/\mathrm{GBSA}}$) of MM-PB/GBSA can be expressed by the following equation:$${\Delta G}_{\mathrm{MM}-\mathrm{PB}/\mathrm{GBSA}}={\Delta G}_{\mathrm{COM}}-{\Delta G}_{\mathrm{PROT}}-{\;\Delta G}_{\mathrm{LIG}}={\Delta E}_{\mathrm{MM}}+{\Delta E}_{\mathrm{sol}}-\;T\Delta S={\;\Delta E}_{\mathrm{vdW}}+{\;\Delta E}_{\mathrm{ELE}}+{\;\Delta E}_{\mathrm{GB}}+{\Delta E}_{\mathrm{SA}}-T\Delta S$$

${\Delta G}_{\mathrm{COM}}$, ${\Delta G}_{\mathrm{PROT}}\;$, and ${\Delta G}_{\mathrm{LIG}}$ stand for the free energy estimation from the protein–ligand complex, protein, and ligand, respectively, in the solvent condition. The ${\Delta E}_{\mathrm{MM}}$ expresses the interaction energy between the protein and ligand in the gas phase, which can be calculated using van der Waals (${\Delta E}_{\mathrm{vdW}}$) and electrostatic interactions (${\Delta E}_{\mathrm{ELE}}$). The ${\Delta G}_{\mathrm{SOL}}$ represents the solvation free energy, which was obtained by calculating the polar solvation (${\Delta E}_{\mathrm{GB}}$) and non-polar solvation (${\Delta E}_{\mathrm{SA}}$) free energy. The TΔS represents the entropy term, which was computed as a more rigorous and concise interaction entropy (IE) proposed by Duan et al. [25] using the same GMX_MMPBSA package.

### 3.4. FPL Simulation

The unbinding pathway was determined using Caver 3.0.3 analysis [26]. The last 1 ns average protein–ligand complex from the MD trajectory was retrieved as the initial structure for the SMD simulation. The protein–ligand complex was placed in a periodic box of 12 Å × 10 Å × 10 Å. The TIP3P water model was used to solvate the system, neutralize with Na^+^ and Cl^−^ counterions, and gradually increase the ion concentration to 0.15 M. The system was then minimized using the steepest descent integrator for 10,000 steps, followed by 200 ps of NPT simulation. The ligand was then pulled from the binding pocket to a distance of about 5 nm. A harmonic force constant of 250 kj mol^−1^ nm^−2^ in the *X*-axis direction was used in the FPL simulation for 500 ps. The pulling speed was maintained at 0.010 nm/ps and the ligand displacement was recorded through the unbinding pathway every 0.1 ps. Each FPL simulation was performed three times to guarantee sufficient sampling. The LIE approximation (${\Delta G}_{\mathrm{LIE}}$) from FPL simulation was calculated according to this study [27]:$${\Delta G}_{\mathrm{LIE}}=\frac{1}{2}\Delta E^{\mathrm{cou}}+\frac{1}{2}\Delta E^{\mathrm{vdW}}$$
where $\Delta E^{\mathrm{cou}}$ = $E_{\mathrm{unbound}}^{\mathrm{cou}}-E_{\mathrm{bound}}^{\mathrm{cou}}$ and $\Delta E^{\mathrm{cou}}$ = $E_{\mathrm{unbound}}^{\mathrm{cou}}-E_{\mathrm{bound}}^{\mathrm{cou}}$ were calculated from the vdW and electrostatic interaction from the *bound* and *unbound* states of the ligand, respectively.

### 3.5. US Simulation

The US process was divided into two stages. In the first stage, the unbinding procedure was performed by SMD/FPL simulation. In the second stage, the initial structures for US simulation were extracted from the SMD trajectories with a spacing distance of 0.2 nm [28]. However, four additional coordinates were assigned for the first 0.8 nm distance, resulting in the first eight windows having a spacing distance of 0.1 nm. A total of 25 protein–ligand conformations from the bound to unbound process were collected as reaction coordinates. After that, each conformation was first equilibrated with a 200 ps NPT ensemble, followed by 2.5 ns of US simulation. The built-in ‘*gmx wham*’ function was used to analyze the potential mean force (PMF) along with its reaction coordinates using the weighted histogram (WHAM) method. Finally, the binding free energy ($\Delta G_{\mathrm{US}}$) was calculated by subtracting the lowest and highest values from the PMF curve. The computational error estimation was carried out by 100 bootstrapping runs.

### 3.6. Free Energy Perturbation

The last 1 ns average MD structures of the receptor–ligand complexes were taken as the initial structure for the FEP simulation study, according to the previous literature [29,30]. In this method, the ligand interaction was turned on over the two-coupling process in the receptor-bound form and isolated form in the solvent. The ligands were transitioned from non-interaction (0) to the full-interaction state (1) by turning on vdW and coulombic interactions with the surroundings by changing the coupling parameter (λ). Nine λ values, 0.00, 0.10, 0.25, 0.35, 0.50, 0.65, 0.75, 0.90, and 1.00, were used to change the vdW and coulombic interactions, and a total of seventeen alter-λ simulations of each 2 ns were performed. The total energy change in the FEP process through the λ alteration was deduced using Bennet’s acceptance ratio (BAR) method.

### 3.7. Dataset Building and Molecular Alignment

The 35 pteridin-7(8*H*)-one-based compounds that were reported to be FLT3 inhibitors were taken as a dataset for this study. The last 1 ns average structure of the C31 from the MD simulation was selected as a representative structure from the dataset. The pteridin-7(8*H*)-one chemical entity was chosen as a common skeleton. Based on the common skeleton, the rest of the compounds were sketched and minimized with a convergence force of 0.05 kcal/mol at the maximum iteration of 2000 run by the tripos force field and added Gasteiger-Hückel partial charges, in the Sybyl suit. The biological activities (IC_50_) of these compounds were converted into logarithmic IC_50_ (pIC_50_) values. The pIC_50_ values were well distributed across the three log units (pIC_50_ = 5.26 to 8.80). The entire dataset was categorized into low, medium, and high activity segments, as described here [31,32]. The 9 compounds were randomly selected from each segment as the test set compounds in such a way that the compounds would cover different activity ranges while maintaining their structural variations. The training set consists of 26 compounds used as a dependent variable (Training set) to construct the 3D-QSAR model, while the 9 compounds were used as the independent variable (Test set) to access the model’s predictive power.

### 3.8. CoMFA and CoMSIA Studies

CoMFA and CoMSIA are two widely popular 3D-QSAR methods. Lennard Jones and Coulombic potential functions were used to compute the steric and electrostatic fields in CoMFA analysis [33,34]. Each compound was placed in a spatial grid one after another during the calculation process with a grid spacing of 2.0 Å. To calculate the structural characteristics of the compounds, each grid space was assigned to the sp^3^ carbon atom with the vdW probe radius of 1.52 Å and a net charge of +1. The energy cut-off of 30 kcal/mol was applied, and the rest of the parameters were set to default. In the CoMSIA model, additional descriptors, such as hydrophobic, H-bond acceptor, and H-bond donor, were also calculated using steric and electrostatic fields. To determine the distance between compound atoms and probe atoms, a Gaussian-type function was applied in CoMSIA with the default attenuation factor(σ) to 0.3.

To produce statistically significant CoMFA and CoMSIA models, the partial least squares (PLS) method was employed to correlate the biological activity and descriptors of the compounds. The leave-one-out (LOO) method was applied in a cross-validation method to obtain the cross-validation coefficient (q^2^), the optimal number of components (ONC), and the standard error of prediction (SEP) by assigning different partial charges. Then, the no validation method was applied to obtain the non-cross-validation coefficient (r^2^), Fisher’s statistics (F value), and standard error of estimation (SEE). In the CoMSIA model, the descriptor fields, such as S, H, E, A, and D, were used in different combinations to produce the best possible statistical model [35]. To examine the internal and external validation of the 3D-QSAR model, we determined the χ^2^, RMSE, MAE, k, k′, |r_0_^2^ − r′_0_^2^|, (r^2^ − r_0_^2^)/r^2^, r_m_^2^, $\bar{r_{m}^{2}}$, Δr_m_^2^, r^2^_pred_, $Q_{F1}^{2}$, $Q_{F2}^{2}$, $Q_{F3}^{2}$, and $Q_{\mathrm{ccc}}^{2}$ matrices as described in the previous literature [36,37,38].

### 3.9. Contour Map Analysis and Design of New Compounds

The field effects from CoMFA and CoMSIA models were presented by 3D StDev*Coeff contour maps with different color schemes. Each contour described the structural characteristic of the compound that could increase or decrease the inhibitory activity. The favorable and unfavorable regions for the steric, electrostatic, hydrophobic, H-bond acceptor, and H-bond donor were colored green, yellow, blue, white, magenta, orange, and cyan. The detailed CoMFA and CoMSIA contour maps were summarized in a simplified SAR scheme. We designed novel compounds based on the MD and SAR and anticipated their inhibitory potency. Compounds with predictive pIC_50_ higher than C31 were subjected to binding affinity evaluation.

## 4. Conclusions

In summary, the overexpression and frequent mutations of FLT3 kinase remain an intriguing challenge in the treatment of AML. We have herein employed the binding studies of pteridin-7(8*H*)-one-based FLT3 inhibitors using docking, MD simulation, and multiple binding energy term calculations. We applied MM-PB/GBSA, FPL, LIE, and FEP scoring functions to correlate the experimental binding energies and computed the binding energies of the selected compounds. In MM-PB/GBSA and FEP methods, the acceptable correlation coefficients ($R_{\mathrm{MM}-\mathrm{PB}/\mathrm{GBSA}}=0.92,\mathrm{and}\;R_{\mathrm{FEP}}=0.71$) were obtained, whereas the rupture force ($F_{\max}$) and LIE are weakly correlated with the experimental binding energies. Although, each binding model distinguished the free energy differences between wild-type and mutant complexes of the most active compounds. In addition, we employed a more rigorous biased sampling (US) simulation to evaluate the effective binding free energies between the FLT3-C31 and FLT3_D835Y_-C31 complexes from the PMF curve. Following that, the statistically significant CoMFA (q^2^ = 0.768, r^2^ = 0.982) and CoMSIA (q^2^ = 0.844, r^2^ = 0.972) models were generated, which showed the strong correlations between the observed and predicted inhibitory activity of the dataset compounds. The developed 3D-QSAR models had a satisfactory predictive power ($r_{\mathrm{pred}}^{2}$ > 0.6) and could be utilized to assess the inhibitory potential of the unknown compounds with analogous scaffolds. We designed several new compounds based on the SAR scheme by growing the chemical entities from the reference molecule C31. Thirteen compounds were predicted to have higher pIC_50_ than the most active compounds and were subjected to binding affinity evaluation by MM-PB/GBSA, LIE, and FEP calculations. Multiple compounds were determined to have greater binding free energy to wild-type and mutant receptors than most active compounds in different scoring models. Overall, these designed compounds have the potential to be lead molecules in future biochemical assays.

## Figures and Tables

**Figure 1 ijms-23-07696-f001:**
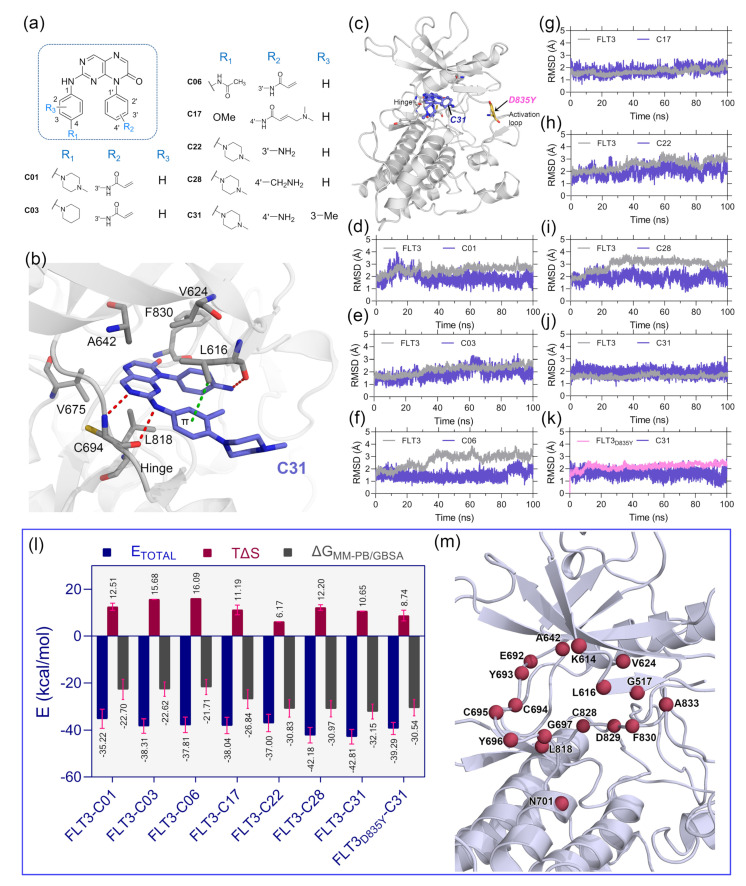
Molecular docking, MD simulation, and MM-PB/GBSA binding energy calculation studies. (**a**) Two-dimensional structure of the pteridin-7(8*H*)-one-based compounds, C01, C03, C06, C17, C22, C28, and C31, which were selected for molecular docking studies. (**b**) Binding pose orientation of C31 within the ATP pocket. The ligand is anchored to the hinge by two H-bond interactions, as shown in red dashed lines, with residue C694. The π–σ interaction with L616 is shown in green dashed lines. (**c**) The global structure of the C31 bound FLT3 kinase domain, depicting the D835Y mutation site in the activation loop. (**d**–**k**) RMSD plots of the selected complexes, i.e., FLT3–C01, FLT3–C03, FLT3–C06, FLT3–C17, FLT3–C22, FLT3–C28, FLT3–C31, and FLT3_D835Y_–C31, respectively. (**l**) A graphical comparison of the $E_{\mathrm{TOTAL}}$, TΔS, and ${\Delta G}_{\mathrm{MM}-\mathrm{PB}/\mathrm{GBSA}}$. The binding energy terms between the complexes are colored blue, firebrick, and gray, with the standard deviation (red). (**m**) Residues that contributed the critical binding energies to the ligands during the per-residue MM-PB/GBSA decomposition analysis are highlighted in the firebrick sphere representation.

**Figure 2 ijms-23-07696-f002:**
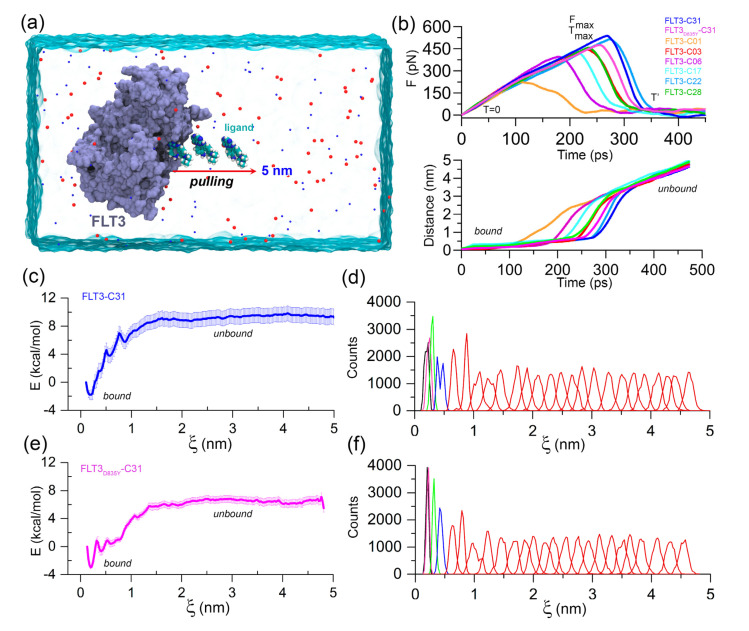
FPL and US simulation analysis. (**a**) Illustration of the FPL simulation setup. An external harmonic force is used to dissociate the ligand to a distance of 5 nm on the *X*-axis. (**b**) Graphical representation of the mean force evaluation (rupture force) and displacement of the ligands with respect to the time during the FPL runs. (**c**) PMF curve and (**d**) histogram profile of the FLT3–C31 system. (**e**) PMF curve and (**f**) histogram profile of the FLT3_D835Y_–C31 system. The errors are calculated by using 1000 bootstrapping runs.

**Figure 3 ijms-23-07696-f003:**
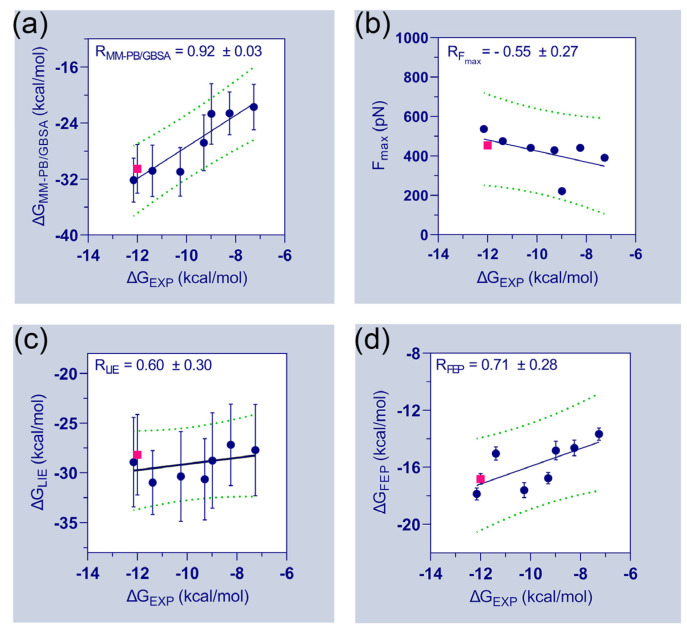
Relationship between experimental and estimated binding energies. In addition to the correlation coefficients of each matrix, the correlation plots of (**a**) $\Delta G_{\mathrm{EXP}}$ vs. ${\Delta G}_{\mathrm{MM}-\mathrm{PB}/\mathrm{GBSA}}$ (**b**) $\Delta G_{\mathrm{EXP}}$ vs. $F_{\max}$ (**c**) $\Delta G_{\mathrm{EXP}}$ vs. ${\Delta G}_{\mathrm{LIE}}$ and (**d**) $\Delta G_{\mathrm{EXP}}$ vs. ${\Delta G}_{\mathrm{FEP}}$ are determined. The experimental and computed binding energies of FLT3_D835Y_–C31 are shown in pink. Standard errors of correlation coefficients are estimated using 1000 bootstrapping runs.

**Figure 4 ijms-23-07696-f004:**
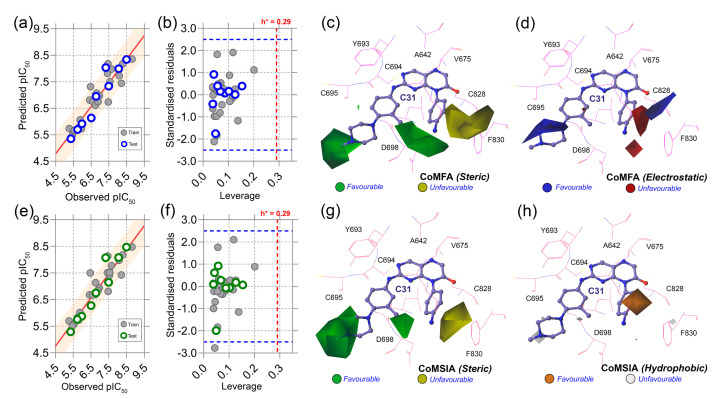
Correlation plots, applicability domain, and contour maps analysis. (**a**) PLS regression plot, (**b**) Williams plot, (**c**) steric contours, and (**d**) electrostatic contours from CoMFA analysis. (**e**) PLS regression plot, (**f**) Williams plot, (**g**) steric contours, and (**h**) hydrophobic contours from CoMSIA analysis. Warning leverage (h*) is shown as red dashed lines in each AD plot. In steric maps, the green and yellow contours indicate that bulky groups are favored and unfavored. On the electrostatic map, blue and red indicate a favorable space for the electropositive and electronegative groups. The orange and light gray color in the hydrophobic map represents favorable and unfavorable substitutions for hydrophobic chemical groups.

**Figure 5 ijms-23-07696-f005:**
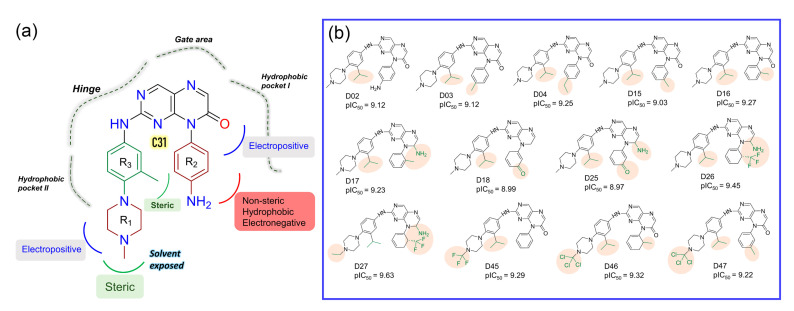
SAR study and development of new compounds. (**a**) The summary of the structure-activity relationship from the CoMFA and CoMSIA analyses around the reference compound C31. (**b**) Designed compounds with higher predicted pIC_50_ in the CoMSIA model. The substitution sites are highlighted in the pale salmon color.

**Figure 6 ijms-23-07696-f006:**
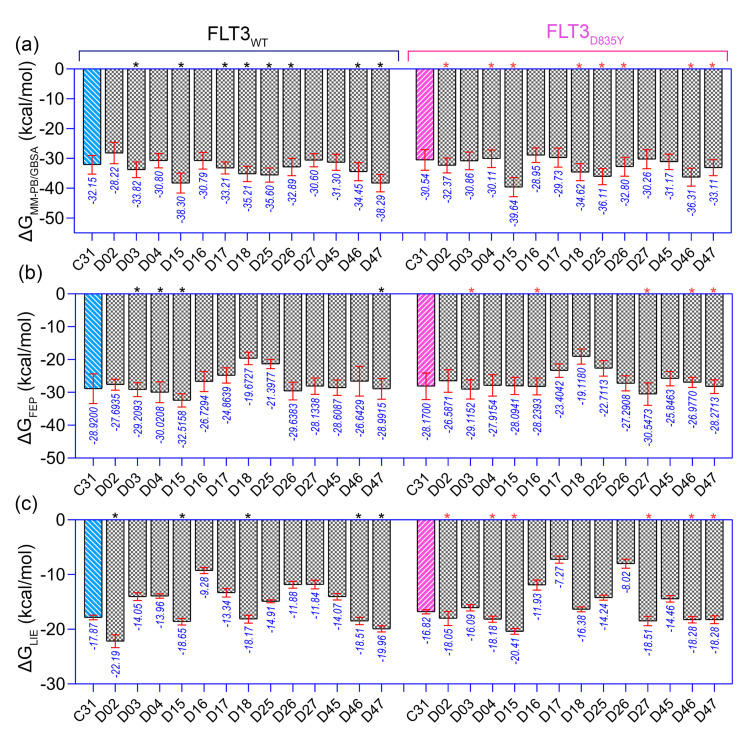
Comparison of the binding free energy of the designed compounds. (**a**) ${\Delta G}_{\mathrm{MM}-\mathrm{PB}/\mathrm{GBSA}}$, (**b**) ${\Delta G}_{\mathrm{LIE}}$, and (**c**) ${\Delta G}_{\mathrm{FEP}}$ binding free energy to wild-type (FLT3) and mutant (FLT3_D835Y_) receptors. The higher estimated binding free energies of the designed compounds than the most active compound C31 are marked by the black (for wild-type FLT3) and pink asterisk (for FLT3_D835Y_). The energy values shown below the standard deviation bar in blue are in kcal/mol.

**Table 1 ijms-23-07696-t001:** Experimental and calculated binding energies from MM-PB/GBSA, FPL, LIE, and FEP calculations. Except for the rupture force ($F_{\max}$), all binding energy terms are expressed in kcal/mol.

**Complexes**	$\Delta G_{EXP}$	$\Delta G_{MM-PB/GBSA}$ $\left(\pm SD\right)$	$F_{max}\;(\mathrm{pN})$	$\Delta G_{LIE}$ (±SD)	$\Delta G_{FEP}$ (±SD)
FLT3–C01	−8.98	−22.70 ± 4.32	221.40	−28.76 ± 4.80	−14.83 ± 0.65
FLT3–C03	−8.24	−22.62 ± 3.06	441.13	−27.18 ± 4.11	−14.64 ± 0.54
FLT3–C06	−7.26	−21.71 ± 3.25	391.61	−28.71 ± 4.61	−13.68 ± 0.43
FLT3–C17	−9.29	−26.84 ± 4.01	428.75	−30.65 ± 4.09	−16.77 ± 0.40
FLT3–C22	−11.38	−30.83 ± 3.68	475.17	−30.97 ± 3.23	−15.04 ± 0.46
FLT3–C28	−10.25	−30.97 ± 3.46	441.13	−30.35 ± 4.51	−17.61 ± 0.53
FLT3–C31	−12.15	−32.15 ± 3.13	537.07	−28.92 ± 4.51	−17.87 ± 0.43
FLT3_D835Y_–C31	−12.00	−30.54 ± 3.47	453.51	−28.17 ± 4.06	−16.82 ± 0.38

**Table 2 ijms-23-07696-t002:** Statistical results and validation of CoMFA and CoMSIA models.

Statistical Parameters	CoMFA	CoMSIA	Threshold Values	Statistical Parameters	CoMFA	CoMSIA	Threshold Values
		SH				SH	
q^2^	0.768	0.844	>0.5	k _Test_	0.989	0.997	0.85 ≤ k or k′ ≤ 1.15
ONC	3	4	<6	k′ _Test_	1.009	1.000	
SEP	0.523	0.439		r^2^ _Test_	0.965	0.931	
r^2^	0.982	0.972	>0.6	$r_{0}^{2}$ _Test_	0.923	0.918	≈r^2^
SEE	0.270	0.194	<<1	${r^{\prime }}_{0}^{2}$ _Test_	0.938	0.929	
F-value	111.678	111.663		$\left|r_{0}^{2\;}-{r^{\prime }}_{0}^{2}\right|$ _Test_	0.009	0.011	<0.3
BS-r^2^	0.959	0.968		$\frac{r^{2}-r_{0}^{2}}{r^{2}}$ _Test_	0.008	0.008	<0.1
BS-SD	0.018	0.016		$\frac{r^{2}-{r^{\prime }}_{0}^{2}}{r^{2}}$ _Test_	0.034	0.034	
χ^2^	0.056	0.066	<1.0	$r_{m}^{2\;}$ _Test_	0.768	0.758	$r_{m}^{2\;}$$\;\mathrm{or}\;{r^{\prime }}_{m}^{2}$ > 0.5
RMSE	0.322	0.352	<0.5	${r^{\prime }}_{m}^{2}$ _Test_	0.790	0.785	
MAE	0.033	0.038	≈ 0	$\Delta r_{m}^{2\;}$	0.022	0.027	
RSS	2.69	3.23		$\bar{r_{m}^{2\;}}$	0.779	0.772	>0.5
k *_Train_*	0.994	0.994	0.85 ≤ k or k′ ≤ 1.15	$r_{\mathrm{pred}}^{2}$	0.919	0.918	>0.6
k′ *_Train_*	1.002	1.002		$Q_{F1}^{2}$	0.919	0.918	
$r_{0}^{2\;}$ _Train_	0.889	0.867	≈r^2^	$Q_{F2}^{2}$	0.919	0.918	
${r^{\prime }}_{0}^{2}$ _Train_	0.877	0.842	≈r^2^ <0.1	$Q_{F3}^{2}$	0.919	0.918	
$\left|r_{0}^{2\;}-{r^{\prime }}_{0}^{2\;}\right|$ _Train_	0.011	0.024		$Q_{\mathrm{ccc}}^{2}$	0.964	0.962	≈1
$\frac{r^{2}-r_{0}^{2}}{r^{2}}$ _Train_	0.011	0.077	<0.3	S (%)	54.2	46.8	
$\frac{r^{2}-{r^{\prime }}_{0}^{2}}{r^{2}}$ _Train_	0.018	0.104	<0.3 $r_{m}^{2\;}$$\;\mathrm{or}\;{r^{\prime }}_{m}^{2\;}$ > 0.5	E (%)	45.8	NA	
$r_{m}^{2\;}$ _Train_	0.683	0.686		H (%)	NA	53.2	
${r^{\prime }}_{m}^{2}$ _Train_	0.664	0.646	$r_{m}^{2\;}$$\;\mathrm{or}\;{r^{\prime }}_{m}^{2}$ > 0.5				

q^2^: squared cross-validated correlation coefficient; ONCs: optimal number of components; SEP: standard error of prediction; r^2^: squared correlation coefficient; SEE: standard error of estimation; F-value: F-test value; BS-r^2^: bootstrapping squared correlation coefficient; BS-SD: standard deviation from 100 bootstrapping runs; χ^2^: chi-squared value; RMSE: root-mean-square error; MAE: mean absolute error; k: slope of the predicted vs. observed activity at zero intercepts; k’: slope of the observed vs. predicted activity at zero intercepts; $r_{0}^{2\;}$: squared correlation coefficient between predicted and observed activity; ${r^{\prime }}_{0}^{2}$: squared correlation coefficient between predicted and observed activity; $r_{m}^{2\;}$, ${r^{\prime }}_{m}^{2}$: $r_{m}^{2\;},{\;\mathrm{and}\;r^{\prime }}_{m}^{2}$ matrices; $\Delta r_{m}^{2\;}=|r_{m}^{2\;}-{r^{\prime }}_{m}^{2}|$; $\bar{r_{m}^{2\;}}=\left(r_{m}^{2\;}+{r^{\prime }}_{m}^{2\;}\right)/2$; $r_{\mathrm{pred}}^{2}$: predictive correlation coefficient; $Q_{F1}^{2}$, $Q_{F2}^{2}$, $Q_{F3}^{2}$, and $Q_{\mathrm{ccc}}^{2}$: $Q_{F1}^{2}$, $Q_{F2}^{2}$, $Q_{F3}^{2}$, and $Q_{\mathrm{ccc}}^{2}$ matrices; S: steric; E: electrostatic; H: hydrophobic; A: H-bond acceptor; D: H-bond donor.

## Data Availability

Data available within the article or its Appendix A.

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
