# Peer review of "Binding Studies and Lead Generation of Pteridin-7(8H)-one Derivatives Targeting FLT3"

_ijms, 2022, doi:10.3390/ijms23147696_

Round 1

Reviewer 1 Report

In the article “Binding studies and lead generation of pteridin-7(8H)-one derivatives targeting FLT3”, the authors design new compounds targeting FLT3 by employing molecular modeling techniques, which could lead to kinase drug development. But there are suggested improvements below in order to corroborate the suitability for publication.

Major concern:

Whether the persuasive power of computer molecular simulations is sufficient, is there any experiments need to be added to prove it?

Minor concern:

Line 104~106, whether the simulation of the missing residues and the subsequent simulations based on the simulation of the missing residues will cause the final result to deviate significantly from the actual situation.

Author Response

We acknowledge the reviewer for valuable feedback on our manuscript. Please see the attached file.

Reviewer 2 Report

The manuscript entitled “Binding studies and lead generation of pteridin-7(8H)-one derivatives targeting FLT3” is a wonderful study and the discussion and methodology are also great.

I have two notes only of the manuscript, I see, they could be helpful for different readers to understand the study much well

The first, the authors mentioned the compounds in the study using symbols such as C31 (for example) the most active compound, the structure is not clear as well in the docking structure, I see if the compounds used in the study are drawn using chemDraw program or chemskech for example to be very clear for the readers, it will be wonderful.

 The second, the chemical structures if named with IUPAC name will be much better, such as, N-phenylpyrimidine-2-amine and N-phenylpyrazine-2-amine line 262, it is better to be 2-phenylaminopyrimidine and 2-phenylaminopyrazine. Also, the letter N if used it must be italic such as N-phenylpyrazine-2-amine

pteridin-7(8H)-one sometimes written with e and another without as in the lines 11, 478. The right form is without and the H must be italic, pteridin-7(8H)-one

Author Response

(The authors gave the same response as above.)

Reviewer 3 Report

The Manuscript ID: ijms-1797232 given to me to review in my opinion is adequate to publish in the IJMS journal. 

The manuscript titled "Binding studies and lead generation of pteridin-7(8H)-one derivatives targeting FLT3" is written by Suparna Ghosh and Seung Joo Cho and its concern is the series of pteridine-7(8H)-one derivatives targeting wild-type FMS-like tyrosine kinase-3 (FLT3) and its D835Y mutant (FL3D835Y) that were studied using a combination of molecular modeling techniques such as docking, molecular dynamics (MD), binding energy calculation, and three-dimensional quantitative structure-activity relationship (3D-QSAR) study.   In the text I found some mistakes or irregularities that should be corrected.   Page 3 line 127: “Adequate amounts of Na+ and CL- were added” should be Cl. Page 4 line 188: “Nine λ values: 0.00, 0.10, 0.025, 0.35, 0.50, 0.65, 0.75, 0.90 were used” but we have only eight values. The whole manuscript needs some attention to eliminate typos like: Page 4 line 195 written like this “as a data set” and like this line 196 “from the dataset”.   Or page 6 line 253 : “An RMSD between 2.0-3.0 Å between the docked pose and crystal pose is an acceptable docking”   Page 7 line 300, 301: “Compounds C01, C03, C06, C17, C22, and C28 were estimated to have the binding energies of -22.70 kcal/mol, -21.71 kcal/mol, -22.62 kcal/mol, -21.71 kcal/mol, -26.84 kcal/mol, -30.38 kcal/mol, and -30.97 kcal/mol, respectively.” In this sentence are 6 compounds but we have seven values.   In supplementary material in Table S6 for compounds 04 and 22 we do not have any substituents marked.   What was the reason to cut the compounds for two sets 26 and 9. How did you select those nine compounds?   Furthermore, in some of study there is chosen a different set of compounds C31, C01, C03, C06, C17, C22 and C28 than in the rest why? What was the reason?

Author Response

(The authors gave the same response as above.)
